# A Novel Effector *FlSp1* Inhibits the Colonization of Endophytic *Fusarium lateritium* and Increases the Resistance to *Ralstonia solanacearum* in Tobacco

**DOI:** 10.3390/jof9050519

**Published:** 2023-04-27

**Authors:** Jianming Huang, Zhangjiang He, Jiankang Wang, Xingping Zha, Qing Xiao, Guihua Liu, Yongjie Li, Jichuan Kang

**Affiliations:** 1Department of Plant Pathology, College of Agriculture, Guizhou University, Guiyang 550000, China; 2Southwest Biomedical Resources of the Ministry of Education, Guizhou University, Guiyang 550000, China

**Keywords:** endophytic fungi, effector, *FlSp1*, ROS, colonization

## Abstract

Effectors are crucial for the interaction between endophytes and their host plants. However, limited attention has been paid to endophyte effectors, with only a few reports published. This work focuses on an effector of *Fusarium lateritium*, namely *FlSp1* (Fusarium-lateritium-Secreted-Protein), a typical unknown secreted protein. The transcription of *FlSp1* was up-regulated after 48 h following fungal inoculation in the host plant, i.e., tobacco. The inactivation of *FlSp1* with the inhibition rate decreasing by 18% (*p* < 0.01) resulted in a remarkable increase in the tolerance of *F. lateritium* to oxidative stress. The transient expression of *FlSp1* stimulated the accumulation of reactive oxygen species (ROS) without causing plant necrosis. In comparison with the wild type of *F*. *lateritium (WT*), the *FlSp1* mutant of the *F. lateritium* plant (Δ*FlSp1*) reduced the ROS accumulation and weakened the plant immune response, which resulted in significantly higher colonization in the host plants. Meanwhile, the resistance of the Δ*FlSp1* plant to the pathogenic *Ralstonia solanacearum*, which causes bacterial wilt, was increased. These results suggest that the novel secreted protein *FlSp1* might act as an immune-triggering effector to limit fungal proliferation by stimulating the plant immune system through ROS accumulation and thus balance the interaction between the endophytic fungi and their host plants.

## 1. Introduction

During growth, plants are confronted with a complex microenvironment in which pathogenic microbes and endophytes are present. In order to be protected from pathogens, plants have evolved two layers of immune systems: pattern-triggered immunity (PTI) and effector-triggered immunity (ETI). In order to avoid the recognition by their host plant, pathogens interfere or disrupt the PTI response by secreting effectors to suppress the plant’s immune system, while plants have evolved corresponding R proteins to specifically recognize effectors, triggering a higher-intensity defense response-ETI [1]. The effectors and plant defense systems evolve simultaneously in a dynamic balance and compete with each other. However, the function of endophytic effectors remains unclear. Therefore, exploring the mechanism of effectors in endophyte–plant interactions plays an important role in revealing the function of endophytes or the symbiotic relationship between endophytes and plants.

ROS burst plays an important role in plant immune systems [2]. For example, via the expression of defense-related genes through the overexpression of the transcription factors DEAXW and matrix metalloproteinases (MMP), both arabidopsis and tomato promote the accumulation of ROS and enhance their resistance to *Botrytis cinerea* [3,4]. In order to counter the defense strategy of ROS burst in the host plant, pathogens have developed new virulence mechanisms to escape plants’ immune systems. The Glomerella leaf spot of apple disease secretes an effector, namely *Sntf2*, to inhibit a plant’s immune system by reducing the accumulation of callose and H_2_O_2_ in the plant and enhancing its own virulence [5]. Additionally, *Puccinia striiformis f.* sp. *tritic* reduced host immunity with the increase in H_2_O_2_ accumulation in wheat and inhibited ROS-mediated cell death by silencing the effector *pstGSRE1* [6]. Pathogenic effectors can also achieve ROS inhibition by activating or mimicking a plant’s ROS scavenging system. The fungal effector *AvrPiz-t* structurally mimicked *ROD1* and activated the plant’s ROS scavenging system to suppress the plant immune system and promote fungi infestation [7]. These results suggest that the mechanism of ROS burst is the “Battlefield” of the competition between effectors and plant immunity. The accumulation of ROS enhances the resistance of plants. Fungi can inhibit the accumulation of ROS in plants, weaken the immune system of plants, and promote colonization and infection. In addition to pathogens, endophytic fungi can also affect ROS in plants [8], but the mechanism is still unclear. Based on the above, it can be concluded that the mechanisms of plant ROS burst play an important role in the plant immune response [9].

In previous studies, *F. lateritium* was usually considered to be a pathogen [10]. In our research, a strain of *F. lateritium* was isolated from *Nothapodytes pittosporoides*, which acted as an endophyte to promote plant growth and mediate disease resistance in the plants of Solanaceae [11,12,13]. However, the potential mechanisms by which *F. lateritium* regulates plant ROS burst and manipulates the plant immune system are still poorly understood. In this study, we investigate the mechanism of *F. lateritium*–plant interactions. An effector, namely *FlSp1*, was screened via bioinformatics analysis, revealing it to be an uncharacterized protein without a functional, structural domain. During the further examination of the function of the effector, we found that *FlSp1* reduced the colonization of fungi and weakened tobacco resistance to *Ralstonia solanacearum* through the regulation of the plant immune system. The results of this study may provide a preliminary theoretical basis for *F. lateritium*–plant interactions and the mechanism by which *F. lateritium* helps plants to resist disease.

## 2. Materials and Methods

### 2.1. Plant Growing Conditions

Tobacco was cultured on solid Murashige and Skoog (MS) medium (0.443% MS Basal Medium with Vitamins; 0.7% agar powder; 3% sucrose; pH = 5.8). The tobacco seeds were washed 5 times with sterile water, then the seeds were soaked in 75% alcohol for 1 min followed by 2% NaOCl for 10 min, and finally, the seeds were washed 5 times with sterile water. The cleaned seeds were placed onto MS medium at 22 °C in a 16 h photoperiod/8 h dark cycle.

To cultivate the tobacco in soil, tobacco seeds were sown in soil and cultured at 28 °C in a 16 h photoperiod/8 h dark cycle and 40–60% humidity. Then, the seeds were transplanted to the pots after germination and cultured for 4 weeks until root irrigation or injection.

### 2.2. Strain Growth Conditions

The wild type and mutants of *F. lateritium* were incubated on potato dextrose agar (PDA) medium (0.5% potato extract; 2.0% glucose; 1.5% agar powder; 0.1 mg/mL chloramphenicol) at 25 °C in the dark for 8 d. The colonies were inoculated into 1/4 Sabouraud dextrose broth (SDB) liquid medium (0.5% yeast extract; 0.25% peptone; 1% glucose) for 5 d at 28 °C in the dark at 160 rpm. The mycelium was separated using triple filter paper and centrifuged for 5 min to obtain the spore precipitate. Finally, the concentration of the precipitate was adjusted to 1 × 10^6^ spores/mL for co-culture or root irrigation.

*R. solanacearum* was inoculated in BGT medium (0.5% yeast extract; 0.1% bacteriological peptone; 0.1% casamino acid; 0.5% glucose; 1.5% agar powder; 0.5% 2,3,5-triphenyltetrazolium chloride) and incubated at 28 °C for 48 h under dark conditions. Additionally, single colonies were placed into B medium (1% bacteriological pepton; 0.1% yeast extract; 0.1% casamino acid) at 28 °C and 180 rmp under dark conditions until OD_600_ = 1.0 for the next experiment.

### 2.3. Bioinformatic Prediction of Effectors in F. lateritium

Referring to the method by [14,15], we performed signal peptide prediction. All protein sequences of *F. lateritium* were analyzed using signalP4.0, a software that predicts the presence of potential signal peptide sequences in amino acid sequences and their cleavage sites. The results obtained by the software consisted of three main values: C, S, and Y. The highest value of C is the signal peptide cut site, while each amino acid corresponds to an S value, with higher S values in the signal peptide region, and the Y value is a parameter derived from the combined analysis of C and S values.

Secreted proteins prediction was performed as follows: transmembrane proteins were predicted using the software TMHMM for proteins containing signal peptides, and proteins containing transmembrane helices were transmembrane proteins. Among all the proteins containing signal peptides, those containing transmembrane structural domains were removed, and those remaining were secreted proteins.

Effector prediction involved the analysis of secreted proteins using the software EffectorP to predict whether they were effector proteins, and a prediction score was given.

For subcellular localization prediction, the subcellular localization of proteins was predicted using Protcomp9.0, and results were scored.

### 2.4. Plasmid Construction

The knockout of the gene was achieved using a homologous recombination strategy. *FlSp1* was amplified from the *F. lateritium* genome and primers were designed using double-joint PCR to amplify upstream and downstream fragments of 800–1200 bp as homologous arms. At the same time, the hygromycin (hyg) sequence (1026 bp) was amplified, as well as its promoter GPDA (752 bp). The amplification primers corresponding to each fragment were designed by homologous recombination using the software Premier6. The upstream and downstream sequences of *FlSp1* were homologously recombined with the GPDA + hyg fragment to replace the sequence after 50 bp of *FlSp1*. The recombined fragment was transformed into the pK_2_Gus vector and stored for backup.

### 2.5. Genetic Transformation

The agrobacterium tumefacien (AGL1)-mediated genetic transformation of *F. lateritium* with reference to Beauveria bassiana [16] was slightly improved. The recombinant plasmid was transformed into AGL1 and inoculated into agrobacterium rhizogene solid medium (YEB) (1% tryptone; 0.1% yeast extract; 0.05% MgSO_4_·7H_2_O; 0.5% sucrose; 1.5% agar powder; pH = 7.0; 50 μg/mL Kana, and 50 μg/mL Car) at 28 °C for 2 d. A single colony was placed into YEB liquid medium and incubated at 28 °C for 20 h at 180 rmp in the dark. Induction medium (IM) (0.3‰ NaCl; 0.3‰ MgSO_4_·7H_2_O; 0.3‰ K_2_HPO_4_; 0.78% MES; pH = 5.3; 10 mmol/mL glucose; 400μm/mL Acetosyringone (AS)) was added to resuspend the bacteria at OD_660_ = 0.15. The bacteria were incubated at 28 °C for about 6 h at 180 rmp in the dark. At the same time, the spore concentration of fungi was adjusted to 1 × 10^4^ spores/mL with IM and set aside. The spore suspension was mixed with an equal volume of AGL1, and the mixture of 100 μL was placed on microporous membrane on IM medium (200 μmol/L AS and 5 mmo/L glucose) and incubated for a total of 48 h at 22 °C. The microporous membrane was transferred to a Czapek medium plate (CZM) (3% sucrose; 0.2% NaNO_3_; 0.05% MgSO_4_·7H_2_O; 0.1% K_2_HPO_4_; 0.0001% FeSO_4_·7H_2_O; 0.05% KCl; pH = 7.0; 1.5% agar powder; containing 200 mg/mL Cephalosporins(Cef); 25 μg/mL hyg) with thaumatin resistance for the next step of transformer verification.

### 2.6. Transient Expression of FlSp1

Referring to the method of [17], the mCherry::*FlSp1* transient expression vector was transformed into GV3101. Additionally, the transformed strain was incubated in YEB liquid medium at 180 rpm for 18 h. The precipitate was obtained by centrifugation for 5 min and resuspended in MES buffer (2.132 g MES, MgCl_2_·6H_2_O, pH = 5.6), and OD_600_ was adjusted to 0.6 and incubated at 180 rpm for 1 h. The Agrobacterium was injected into 5–6-week-old *Nicotiana benthamiana* with a 1 mm needle-free syringe. After 1 day, the leaves were stained by DAB to detect the ROS burst degree of the leaves. After 3–5 days, the incidence and the cell death area of the leaves were measured. INF1, which is a necrosis-inducing effector of a pathogen, was used as a positive control.

### 2.7. Phenotypic Analysis

The spore concentration of the WT or Δ*FlSp1* was adjusted to 5 × 10^5^ spores/mL using Tween80, and 1 μL of spores suspension was inoculated onto CZM medium containing different stress factors (5.76 mM H_2_O_2_, pH = 10.4, pH = 4, 25 μg/mL Congo Red), respectively. CZM was used as base medium, and after 7–8 d of incubation at 25 °C under dark conditions, the diameter of each colony was measured using digital calipers and fungal growth inhibition rate statistics were performed. The inhibition rate of mycelial growth was calculated using the following equation: inhibition rate of mycelial growth (MGI) (%) = [(RGR − rgr)/RGR] × 100%. RGR is the radial growth rate of the control group, and rgr is the radial growth of fungi.

### 2.8. F. lateritium Co-Cultured with Tobacco

The spore concentration of *fungi* was adjusted to 5 × 10^5^ spores/mL. The sterile 4–6 week-old tobacco was placed in the spore suspension and co-cultured in an incubator at 25 °C with 16 h of light per day, or spore suspension was used to irrigate tobacco roots.

### 2.9. R. solanacearum Inoculation Treatment of Tobacco

*R. solanacearum* was activated according to the method in Section 2.2, the precipitate was collected by centrifugation at 6000 rmp/min for 10 min, and the concentration was adjusted to 1 × 10^8^ spores/mL with sterile water for inoculation. A needleless syringe was used to inject 200 μL of bacterial solution into the abaxial surface of the leaves. The injected parts of the leaves were cut off after 2 d, and the expression of defense-related genes was analyzed by RT-qPCR. After 14 days post-injection, the disease was observed according to disease-grading criteria. Six replicates were set up for each group of treatments. There are five disease severity ratings of the typical symptoms of bacterial wilt, ranging from 0 to 4 [18]: 0 = no symptoms; 1 = 1/4 inoculated leaves wilted; 2 = 1/4–1/2 inoculated leaves wilted; 3 = 1/2–3/4 inoculated leaves wilted; and 4 = whole plant wilted. The disease index was calculated according to the formula D = Σ(Mi × Si)/N, where D is the condition index, i is the number of disease stages, Mi is the number of strains with disease i, Si is the value of stage with disease i, and N is the total number of strains.

### 2.10. Histological Staining of Tobacco

Referring to the slightly improved method of Jing [19], the co-cultured tobacco roots were placed in 0.5 mM NBT solution (0.5 g NBT dissolved in 50 mm/L, pH = 7.8 sodium phosphate buffer, fixed to 500 mL), incubated in the dark for 1 h, washed three times with sterile water, and photographed under a fluorescent microscope.

Referring to the slightly improved method of L. Zhang [20], 0.1 g of DAB was dissolved in 100 mL sterile water, the pH was adjusted to 3.8 using HCl, and the mixture was kept away from light. The leaves were placed in DAB solution and stained overnight in the dark. Then, the leaves were placed in a decolorizing solution (anhydrous ethanol:propanetriol:lactic acid = 3:1:1), boiled for a few minutes until the green color of the leaves had completely faded, and then observed and photographed.

The method of *F. lateritium* mycelium staining was taken from [21]: tobacco roots were washed three times with sterile water after co-culturing, stained in solution (10 μg/mL WGA488, 15 μg/mL PI) for 40 min in the dark, and washed three times with 1 × PBS buffer. Finally, samples were analyzed microscopically on glass slides in 20% glycerol [22].

### 2.11. RNA Extraction and RT-qPCR

The collected samples were washed with sterile water and then snap-frozen. Total plant RNA was extracted using CWbio’s RNApure plant kit; the RNA was reverse-transcribed using the StarScript Ⅱ First-strand cDNA Synthesis Kit, and the RNA was extracted using SYBR^®^ Green qPCR Mix (Monad) on a gradient fluorescent quantitative PCR system (RT-qPCR was performed on a Bio-Rad system using SYBR^®^ Green qPCR Mix (Monad)).

### 2.12. Data Processing

The results of this study are expressed as mean values with standard deviation. The significance of the difference was determined via a one-way ANOVA test with Tukey’s multiple comparison method or t-test with SPSS [23]. Image processing was performed using GraphPad prism version 8.0.

## 3. Results

### 3.1. Prediction of F. lateritium Effectors

In order to investigate the mechanism of *F. lateritium*–plant interactions, we used bioinformatics strategies to predict the effectors of *F. lateritium* and to screen candidate proteins for further study. Firstly, the protein sequences were analyzed by signalP4.0. Additionally, it was found that among 14,756 protein sequences, there were 1502 amino acid sequences containing signal peptides, accounting for 10.17% of the total gene sequences. Further analysis of the proteins containing signal peptides using TMHMM and EffectorP revealed 1139 total secreted proteins and predicted 213 effectors, representing 7.7% and 1.4% of the total gene sequences, respectively (Appendix A). The BLAST comparison revealed that 69 effectors were characterized as proteins, accounting for 32.4% of the total number of effectors (Appendix A). As the number of the amino acid residues of the effectors mostly ranged from 50 to 300 aa, we excluded proteins with amino acid residue numbers greater than 300 aa. Finally, based on the predicted values returned by the software for each protein, we listed the 16 effectors with the highest predicted values (Table 1).

We used the gene with the highest predicted value, EV0011051.1, as the study object. A protein was found to contain 107 amino acid residues and an exocytotic signal structure at the N-terminal end with the signal peptide positioned at 1–17 aa, which had the highest effector likelihood score—0.979—for a typical unknown secreted protein, which was named as *FlSp1* (Fusarium-lateritium-Secreted-Protein). Additionally, the subcellular localization of *FlSp1* was predicted using Protcomp 9.0. Its secretion to the extracellular compartment had the highest predicted value of 2.4, which indicated that *FlSp1* was most likely to be secreted to the extracellular compartment.

### 3.2. FlSp1 Negatively Mediates the Oxidative Stress Response in F. lateritium

To investigate the biological function of *FlSp1*, we obtained a gene mutant of *FlSp1* using a homologous recombination strategy (Appendix A). Phenotypic analysis revealed no significant difference in the growth of the wild type (WT) and the *FlSp1* mutant (Δ*FlSp1*) in normal basal medium (Figure 1A). Additionally, the knockout of *Flsp1* did not affect the growth of the strain under acid-base conditions (pH = 4 and pH = 10.4) and cytostatic (Congo red) conditions. However, under oxidative stress (H_2_O_2_) conditions, the knockout of *Flsp1* resulted in a significant increase in fungal tolerance to oxidative (Figure 1A,B) with relative inhibition rates of 18% (*p* < 0.01). Further analysis revealed that the knockout of *Flsp1* resulted in a significant elevation (~1.25 fold) of peroxidase (CAT) activity in the strain in comparison with WT (*p* < 0.01) (Figure 1C). The analysis of transcriptional patterns revealed that the disruption of *Flsp1* caused the significant up-regulation (~3.6- and 9.1-fold) of the expression of CAT synthesis-related genes—Flcat1 and Flcat2 (*p* < 0.01) (Figure 1D). This suggests that *Flsp1* negatively mediates the oxidative stress responses in *F. lateritium*.

### 3.3. FlSp1 Stimulates Plant Immune Defense Response and Accumulates ROS

To further investigate the biological function of *FlSp1* during fungal–plant interaction, the expression pattern of *FlSp1* was analyzed during the interaction. The results showed that *FlSp1* maintained low expression at 12 and 24 h of the interaction, reached peak expression at 48 h, and started to decrease at 72 h (Figure 2A), indicating that *FlSp1* acts during the early stage of the interaction. Thus, we speculate that *FlSp1* might be involved in the regulation of the plant immune systems during infestation with fungi. To test this theory, we examined the ability of *FlSp1* to cause cell death or ROS accumulation via transient expression. The results showed that both the experimental group with a signal peptide (*FlSp1*^+sp^) and that without a signal peptide (*FlSp1*^−sp^) were able to cause the accumulation of ROS in comparison with the negative control (Figure 2B). In addition, we found that *FlSp1* was unable to cause cell death in plant leaves, regardless of whether the signal peptide was removed (Appendix A). This result indicates that *FlSp1*, although it is able to stimulate plant immunity and ROS accumulation, does not cause a hypersensitive response to plant damage.

Therefore, we presume that *FlSp1* may play a similar role in the interaction between *F. lateritium* and the host plant. We detected the early defense marker—O_2_—of the ROS through NBT staining. The results showed that the Δ*FlSp1*-treated roots produced significantly less blue insoluble material compared to the wild type (Figure 2C), which was examined for grayscale values, and it was found that the degree of ROS burst was reduced by approximately 50% (Figure 2D). Additionally, the significantly lower expression (~18.5 fold) of *ACRE31*, a marker gene for the plant immune system (*p* < 0.001), was detected in comparison with the wild type (Figure 2E,F). The expression of *WRKY8*, a transcription factor that positively regulates plant immune defense, was significantly down-regulated (~4.47 fold) (*p* < 0.001) (Figure 2F). These results suggest that *FlSp1* triggers the accumulation of ROS in plants and mediates the immune system response of plants via the up-regulation of defense-related genes.

### 3.4. Disruption of FlSp1 Enhances the Colonization Rate of F. lateritium in Plants

The level of endophytic fungi colonization in plants is related to their function. To investigate the biological function of *FlSp1* during interactions, we examined the effect of *FlSp1* on fungal colonization in plant roots. The results revealed that the knockout of *FlSp1* resulted in a significant increase in the fungal colonization of plants (Figure 3A), and the colonization of Δ*FlSp1* in the root system was 3.3-fold higher than that of WT. (*p* < 0.01) (Figure 3B). It was shown that *FlSp1* negatively regulated fungal colonization in plants.

### 3.5. FlSp1 Negatively Mediates the Resistance of Tobacco to R. solanacearum

To investigate the effect of *FlSp1* on the *F. lateritium*-mediated promotion of growth and disease resistance in the host plant, we first investigated the effect of *FlSp1* on the growth-promoting effect of *F. lateritium*. Tobacco roots were irrigated with wild type and Δ*FlSp1* spore suspension, and water was used as a control (Appendix A). After 14 d, we counted the fresh weight, plant height, and chlorophyll content of tobacco leaves. The results showed that there was no significant effect on plant growth through *FlSp1* knockdown (Appendix A). To further investigate the effect of *FlSp1* on plant resistance to disease, we used the spore suspensions of wild type and Δ*FlSp1* to irrigate the tobacco roots. After 3 days, the plants were injected with *R. solanacearum,* and the plant disease index was created after 14 d. The results showed that Δ*FlSp1* reduced the disease incidence of the plants compared with the wild type (Figure 4A). The disease indexes for the control, wild-type, and mutant groups were 3.0, 2.4, and 1.5, respectively. The results show that the Δ*FlSp1* group had a significantly lower index of disease (Figure 4B). Meanwhile, *ERF1*, a key factor of the ethylene signaling pathway, and *PR1a*, a pathogenesis-related protein, were significantly up-regulated (~1.7- and 8.7-fold), while *WRKY22*, a transcription factor that negatively regulates plant immune defense, was significantly down-regulated (~7.2 fold) 2 d post-*R. solanacearum* injection (Figure 4C–E). The above results suggest that *FlSp1* regulates plant resistance to *R. solanacearum* by mediating the expression of defense-related genes.

## 4. Discussion

In previous studies, *F. lateritium* has often been reported as a pathogenic fungus [10]. In our research group, a strain of *F. lateritium* was isolated from *Nothapodytes pittosporoides*, which acted as an endophyte to promote plant growth and mediate disease resistance in the plants of Solanaceae [11,12,13]. However, how the fungus mediates disease resistance in plants has not yet been reported. In this study, we focused on effectors, regarding them as an entry point, and predicted the effectors of *F. lateritium* through bioinformatics based on the *F. lateritium* genome. A secreted protein, *FlSp1*, which had the highest effector prediction score, was used as the study object. Subsequently, we analyzed the function of *FlSp1* in fungal–plant interactions. Our results suggest that *FlSp1* may act as an elicitor of plant defense responses, which stimulates plant immune responses (including ROS bursts) to limit the fungal colonization in plants. The reduction in *fungal* colonization helps to maintain balance with plants, so that *F. lateritium* does not damage plant tissue as much as the pathogenic fungus-causing disease.

During the process of fungal infection, ROS bursts caused by plant immune responses can inhibit fungal infection [25]. If fungi can better adapt to oxidative stress, it means that the colonization will be greatly increased [7]. Our data show that *FlSp1* is highly expressed in the early stages of fungus–plant interaction, and the destruction of *FlSp1* leads to a significant increase in the strain’s tolerance to oxidative stress. Therefore, we speculated that the *FlSp1* mutant might be more adaptable to ROS bursts in the early interaction stages. The experimental results are consistent with our prediction that the colonization of Δ*FlSp1* on tobacco significantly increased. However, it is interesting that the increase in the colonization of *F. lateritium* does not cause plant disease but rather enhances plant disease resistance. To clarify the reason for this, we initiated the transient expression of *FlSp1*. The results showed that *FlSp1* could cause ROS bursts in plants but not hypersensitive responses. Further experimental data showed that the destruction of *FlSp1* resulted in a significant decrease in the expression of defense-related genes. Those results indicate that *FlSp1* may act as an elicitor of the plant defense response. *FlSp1* is different from the elicitors of the pathogens that have been studied. For example, *FoEG1* from *Fusarium oxysporum* can cause the cell death of *N. benthamiana* leaves, induce the accumulation of ROS, and enhance the resistance of tobacco to *Botrytis cinerea* [20]. The effector *PvRxLR16* from *Plasmopara viticola* can cause cell death and the accumulation of ROS in *N. benthamiana* leaves and enhance the resistance to *Phytophthora capsici* [17]. It has a common characteristic that can trigger a strong immune response in *N. benthamiana*, leading to leaf necrosis and the accumulation of ROS, while *FlSp1* does not cause necrosis. Therefore, we speculate that *FlSp1*, in addition to acting as an elicitor, may also have an important role in maintaining the endophytic relationship and the balance of the plant immune response, thus explaining the increase in Δ*FlSp1* colonization. The growth of endophytes in the root system must be restricted; otherwise, there is the risk and possibility of causing disease [26]. However, why does an increase in the colonization of *F. lateritium* not cause disease? The reason may be that the level of colonization has not yet reached a critical value or that there are other factors that restrict *F. lateritium* colonization, keeping it within a certain colonization range. Our experimental data also showed that resistance to the pathogen was significantly higher in the Δ*FlSp1*-treated group after inoculation with *R. solanacearum.* Additionally, in this process, the expression of *PR1a* and *ERF1*, the marker genes of salicylic acid (SA)- and jasmonic acid (JA)-mediated signaling pathways, was significantly up-regulated. The results indicate that *FlSp1* may indirectly influence the SA and JA signaling pathways in mediating resistance to R. *solanacearum*. However, this seems to contradict the result that Δ*FlSp1* down-regulates plant defense genes. However, it is clear that the low immunity of the endophytes interacting with the plant does not affect the elevated immune defense response of the plant [27,28,29]. Therefore, we speculate that, within a certain range of colonization, there is a positive correlation between the enhancement of the plant immune response by endophytic *F. lateritium.*

## Figures and Tables

**Figure 1 jof-09-00519-f001:**
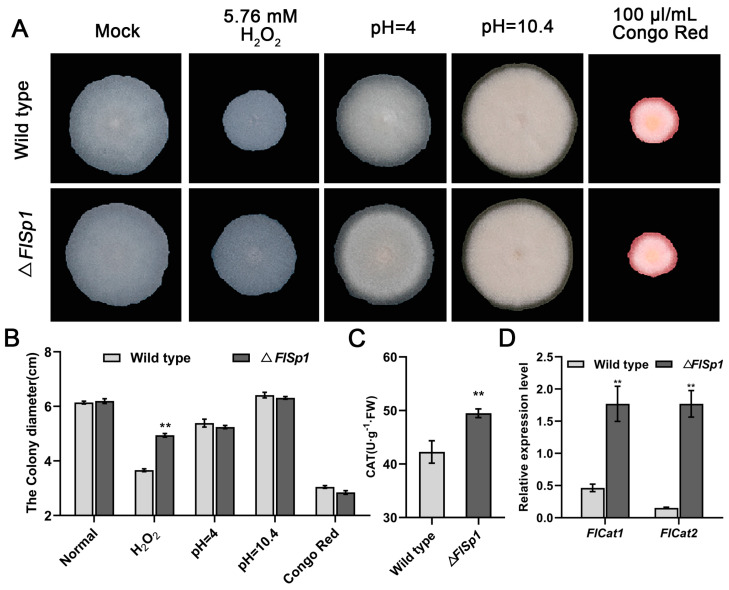
*FlSp1* negatively mediates the oxidative stress response in *F. lateritium*. (**A**) Effect of *FlSp1* on the growth of *F. lateritium* under various stresses. (**B**) The diameter of the colonies was measured. Three replicates per treatment. (**C**) CAT enzyme biopsy assay of mycelium. The same mass was chosen from both wild type and Δ*FlSp1* mycelium and tested for CAT enzyme activity under oxidative stress. (**D**) Analysis of the transcriptional patterns of *Flcat1* and *Flcat2*. Three replicates were set up for each experiment; ** indicates highly significant difference (*p* < 0.01).

**Figure 2 jof-09-00519-f002:**
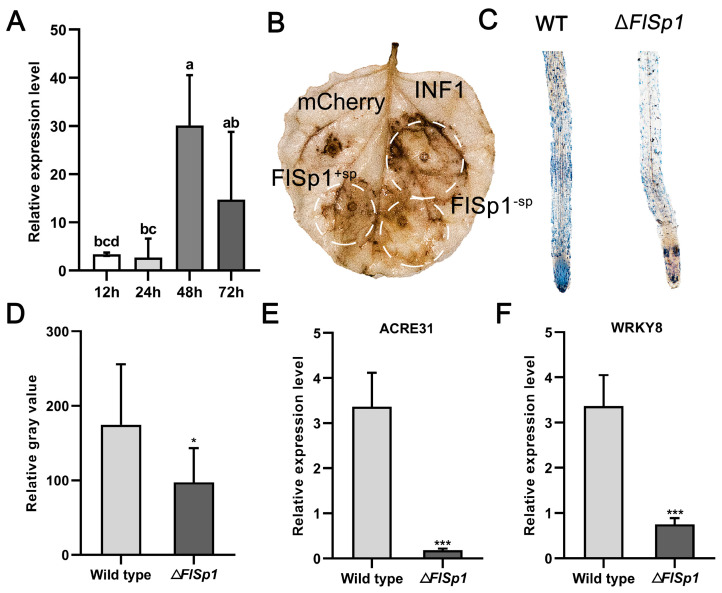
*FlSp1* stimulates the plant immune system and accumulates ROS. (**A**) Expression pattern analysis of *FlSp1*. *FlSp1* is highly expressed in the early stages of interaction. Separate one-way ANOVAs were used for each group, followed by pairwise comparisons at different time points using Tukey’s HSD method with different lowercase letters indicating significant differences in pairwise comparisons (*p* < 0.05). (**B**) *FlSp1^−^*^sp^ and *FlSp1^−^*^sp^ cause ROS accumulation in plant leaves. INF1 was used as a positive control. (**C**) Δ*FlSp1* caused lower ROS burst in plant roots than in wild type. (**D**) Roots were stained for NBT and then examined for grey values using ImageJ. (**E**,**F**) Wild type and Δ*FlSp1* were detected for the relative expression of defense-related genes after 1 d of co-culturing with *N. benthamiana*. Each experiment was repeated three times; * indicates significant difference (*p* < 0.05) and *** indicates extremely significant difference (*p* < 0.001).

**Figure 3 jof-09-00519-f003:**
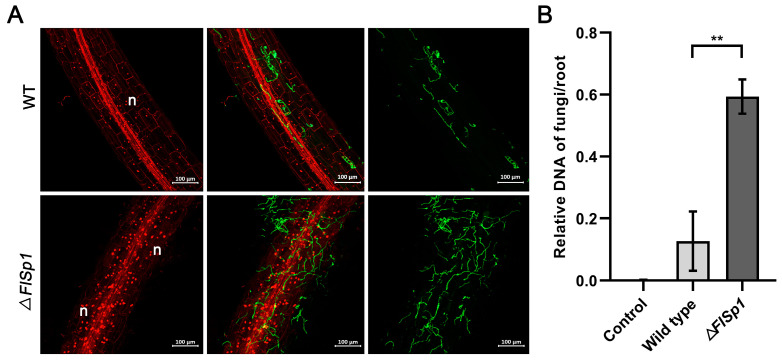
Knockout of *FlSp1* increases fungal colonization. (**A**) The mycelium was stained after 2 d of co-culturing and was photographed at 488 mM. n indicates plant cell nuclei. (**B**) After 2 d of co-culturing, the levels of wild types and mutant DNA were examined in the root system of the plant. Each experiment was repeated three times; ** indicates highly significant differences (*p* < 0.01).

**Figure 4 jof-09-00519-f004:**
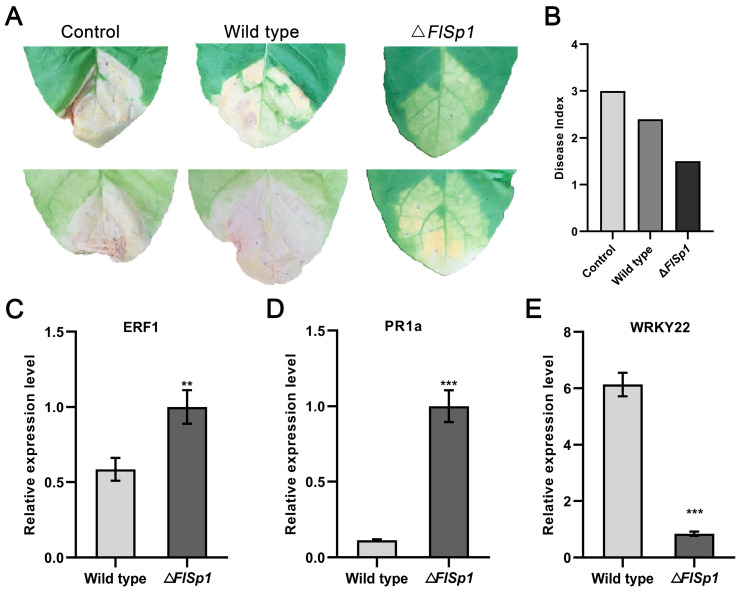
*FlSp1* negatively mediates the resistance of tobacco to *R. solanacearum*. (**A**) The Δ*FlSp1* group had a significantly lower index of disease. (**B**) Disease index for each treatment group. (**C**–**E**) After root irrigation treatment, detection of expression of defense-related genes 2 d after the *R. solanacearum* injection. Each experiment was repeated three times; ** indicates significant difference (*p* < 0.01) and *** indicates a highly significant difference (*p* < 0.001).

**Table 1 jof-09-00519-t001:** Candidate proteins with the highest likelihood scores for 16 effectors. EffectorP predicts fungal effectors based on machine learning. Features that discriminate fungal effectors from secreted non-effectors are predominantly sequence length, molecular weight, and protein net charge, as well as cysteine, serine, and tryptophan content [24].

GeneID	Effector Probability	(aa) Position of Signal Peptide	(aa) Size of Peptide	Subcellular Localization	Integral Prediction of the Protein Location
EVM0011051.1	0.979	1–17	108	Extracellular	2.4
EVM0014202.1	0.971	1–23	127	Extracellular	2.4
EVM0003861.1	0.966	1–16	94	Extracellular	2.3
EVM0007002.1	0.965	1–21	97	Extracellular	2.4
EVM0005253.1	0.959	1–18	121	Extracellular	3.2
EVM0002480.1	0.952	1–17	96	Extracellular	2.5
EVM0001443.1	0.952	1–18	97	Extracellular	2.4
EVM0000104.1	0.952	1–18	89	Extracellular	2.4
EVM0005809.1	0.949	1–18	124	Extracellular	2.6
EVM0004957.1	0.948	1–17	98	Extracellular	2.6
EVM0001142.1	0.943	1–19	88	Extracellular	2.7
EVM0005907.1	0.928	1–17	105	Extracellular	2.7
EVM0013376.1	0.912	1–23	131	Extracellular	2.4
EVM0007040.1	0.91	1–17	65	Extracellular	2.4
EVM0011395.1	0.909	1–16	148	Extracellular	2.4
EVM0001445.1	0.909	1–16	140	Extracellular	2.8

## Data Availability

Not applicable.

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
