# Peer review of "A Novel Effector FlSp1 Inhibits the Colonization of Endophytic Fusarium lateritium and Increases the Resistance to Ralstonia solanacearum in Tobacco"

_jof, 2023, doi:10.3390/jof9050519_

Round 1

Reviewer 1 Report

The study characterized FlSP1, a secreted protein from Fusarium lateritium. The transcription of FlSp1 was found to be significantly down-regulated under oxidative stress conditions but up-regulated during the ring infection process. The authors demonstrated that deletion of Flsp1 in F. lateritium led to increased tolerance to oxidative stress, while transient expression of FlSp1 stimulated the accumulation of reactive oxygen species (ROS) in plants without causing necrosis. Compared to the wild type, ΔFlSp1 plants showed reduced ROS accumulation and weakened plant immune response, leading to increased fungal colonization in host plants. Interestingly, the ΔFlSp1 plant exhibited increased resistance to Ralstonia solanacearum, the pathogen causing bacterial wilt. Overall, the study provides valid results with high-quality presentation. However, the rationale for choosing FlSP1 for further study is not explicitly stated and could be further elaborated in the full manuscript.

Author Response

Dear Reviewers,

Thank you very much for taking the time to review the manuscript and for your very inspiring comments on its merits. All the opinions you give are valuable and very helpful for revising and improving our paper, as well as the important guiding significance to our researches. We have studied comments carefully and have made correction which we hope meet with approval. We start by retyping your comment in black font to facilitate discussion, and then present our response to the comment in red font.

Yours sincerely, Jianming Huang!

comment1: The rationale for choosing FlSP1 for further study is not explicitly stated and could be further elaborated in the full manuscript.

Response1:

We used bioinformatics strategies to predict the effectors of F. lateritium. When we input secreted proteins into SignalP, the software returns the value of the likelihood of the effector for each protein. We used the gene with the highest predicted value, EV0011051.1, as the study object. It was named as FlSp1.

We have described this issue in the manuscript, and we have provided additional explanations(Mainly in parts 3-1 and 2-3). The document submitted is my revised manuscript. 

Thank you again for your encouragement!

Reviewer 2 Report

Authors present functional evidences for involvement of a endophytic effector from Fusarium lateritium. Although it is an outstanding research and authors utilized different sound strategies to prove the research hypotheses, they failed to present their results in coherent way. The discussion is probably the weakest part of manuscript and it is very general and does not explain the details of the studies, if they prove or reject hypotheses and and if they are backed up by the previous research. Several parts of the manuscript needs improvement in English. As an attachment, please find some comments which might help authors to improve the manuscript.

Author Response

Dear Reviewer 2,

Thank you very much for taking the time to review the manuscript. I'm very sorry that there are many errors in my manuscript. All the opinions you give are valuable and very helpful for revising and improving our paper, as well as the important guiding significance to our researches. We have studied comments carefully and have made correction which we hope meet with approval. We start by retyping your comment in black font to facilitate discussion, and then present our response to the comment in red font.

Yours sincerely, Jianming Huang

Line 20-21- Clarification is indeed since it reads as Flsp1 is a plant’s not a pathogen’s gene. The definition of WT vs. ΔFlSp1 is not clearly presented.

I have modified it in the manuscript.

Line 31- The statement of “However, the function of effector remains unclear.” Is wrong as there are increasing list of effectors being functionally characterized. May be authors mean the function of endophytic effectors?

Modified to “Endophytic effector” in the manuscript.

Line 49- “toinhibit” to “to inhibit”. There are numerous cases of extra space character too in other parts of the manuscript. Authors are recommended to use the MS Word proofing before next submission.

I have modified it in the manuscript.

Line 50- accumulate   ion??

Sorry, I have modified it in the manuscript.

Line 51- Tritici to tritici

I have modified it in the manuscript.

Line 52. Increasement and decreasement have been used numerously while they could be replaced with increase and decrease. The decreasement is process of increase and not increase itself. That could come to the fact that authors need to use English improvement services.

I have modified them in the manuscript.

Line 61-In addition to pathogens, endophytic fungi also cause similar reactions (reference).

I have modified them in the manuscript.

Line 63- It needs rewording. What is maintaining plant. It is not a scientific way to consider plant as an engine that needs maintenance. Please clarify by rewording what you mean and use scientific words.

I have modified it in the manuscript.

Line 67- …e resistance in the plants of Solanaceae (reference)

I have modified it in the manuscript.

Line 70- couled must be replaced with could.

I have modified them in the manuscript.

Line 70- “FlSp1 that could cause ROS burst via bioinformatics analysis”, how bioinformatics can predict the role of effectors in fungi? These have been rejected by many other authors. Fungal effectors do not have typical structure or motifs.

I have revised it in the manuscript.

“An effector viz FlSp1 via bioinformatics analysis was screened, which is an uncharacterised protein without functional structural domain. ”

Line 73- “FlSp1 mediated the colonization of fungi”, require rewording. It is vague.

I have modified it in the manuscript.

“we found that FlSp1 reduced the colonization of fungi and weakened tobacco resistance to Ralstonia solanacearum via the regulation of the plant immune system.”

Line 80- “sterilized water, soaked in 75 % alcohol for 1 min, 2 % NaOCl soaked seeds for 10 min,” ia grammatically  wrong.

I have modified it in the manuscript.

“Culture tobacco on solid Murashige and Skoog(MS)(0.443% MS Basal Medium with Vitamins; 0.7% agar powder; 3% sucrose; pH=5.8;) medium: the tobacco seeds were washed 5 times with sterile water, then the seeds were soaked in 75% alcohol for 1 min followed by 2% NaOCl for 10 min, and finally the seeds were washed 5 times with sterile water. The cleaned seeds were picked onto MS medium at 22℃ with 16 h photoperiod/8 h dark cycle. ”

Line 81- Picked is a wrong verb here.

I have modified it in the manuscript.

“The cleaned seeds were picked onto MS medium at 22℃ with 16 h photoperiod/8 h dark cycle.”

Line 83- what is nutrient soil? How were seeds cultured?

I have modified it in the manuscript.

“Cultivating tobacco in soil: Tobacco seeds were sown in soil, cultured at 28 ℃ with 16 h photoperiod/8 h dark cycle and 40% - 60% humidity. Then, the seeds were transplanted to the pots after germination, cultured to 4 weeks for root irrigation or injection.”

Line 90- The mycelium was filtered through triple filter paper; I assume that mycelium was separated, and spore were collected. It requires rewording to reflect that .

I have modified it in the manuscript.

“ The mycelium was separated using triple filter paper and centrifuged for 5 min to obtain the spore precipitate”

Line 91- Finally, the concentration of the precipitate was adjusted to, I assume that concentration of spore was adjusted at certain number of spores per ml. Please report the concentration.

I have modified it in the manuscript.

“Finally, the concentration of the precipitate was adjusted 1 × 106 spores/mL for co-culture or root irrigation”

Line 93- what is BGT medium. Please elaborate.

I have modified it in the manuscript.

Line 107-  those containing transmembrane structural domains, how they were distinguished from others. Was this achieved using software detecting GP-anchored proteins. If so, please cite the software.

This study did not detect GP-anchor protein.

Line 126- YCK is an abbreviation of media not introduced before. Please spell it out and provide the abbreviations in paratheses. Same for line 135 for CZM media.

I have modified it in the manuscript.

Line 144- Nicotiana benthamiana must be italicized.

I have modified them in the manuscript.

Line 145-146- After 3-5 days, the incidence and the cell death area of leaf were counted. Needs rewording as cell death area is measurable not countable.

I have modified it in the manuscript.

“After 3-5 days, the incidence and the cell death area of leaf were measured.”

Line 149- and 1 μL of spores from the wild type and FlSp1 mutant strains were aspirated, what aspiration means here?

I have modified it in the manuscript.

“The spore concentration of the WT or ΔFlSp1 was adjusted to 5 × 105 spores/mL using Tween80, and 1 μL of spores suspension were inoculated onto CZM medium containing different stress factors”

Line 153-156-The inhibition rate of mycelial growth was calculated using the following equation. The inhibition rate of mycelial growth(MGI)(%)=[(RGR-rgr)/ RGR]×100,rgr is the radial growth of F. lateritium,RGR is the radial growth rate of the control. What is RGR-rgr vs RGR?

I have modified it in the manuscript.

“The inhibition rate of mycelial growth(MGI)(%)=[(RGR - rgr) / RGR] × 100%. RGR is the radial growth rate of the control group, and rgr is the radial growth of F. lateritium.”

Line 158- 5 CFU/mL why CFU and not spore per ml?

All "CFU/ML" have been modified.

Line 159- or the spore suspension was used to root tobacco that had been cultured in soil for 4 weeks; need rewording. What is rooting tobacoo?

I have modified it in the manuscript.

Line 167- The injected parts of the leaves were cut off, RT-qPCR analysis was performed after 2 d. It is not clear what hypothesis has been tested using RT-qPCR.

I have modified it in the manuscript.

“and the expression of defence-related genes was analysed by RT-qPCR.”

Line 168- What is disease grading criteria, if this is a disease evaluation scale, it must be clearly stated, or a reference must be provided for the method.

I have modified it in the manuscript.

“There are five disease severity ratings of typical symptoms of bacterial wilt from 0 to 4[14]: 0 = no symptoms, 1 = 1/4 inoculated leaves wilted, 2 = 1/4 - 1/2 inoculated leaves wilted, 3 = 1/2 - 3/4 inoculated leaves wilted, 4 = whole plant wilted. “

Line 177-179- Please consider removing, technical aspect of staining that could be read in the historical references and does not add any value to the content of this work. Similarly lines 184-186 and 192-195.

I have deleted this section.

Line 210- Reference for SPSS.

I have modified it in the manuscript.

Line 213-215- It seems to be part of Instruction for Author document.

Sorry, I have deleted this section.

Line 228- What does highest predicted values mean? How was the value given on these 16 effectors?

When the sequence of my protein is entered into the software, the software will return the predicted effector values for each protein. And this has been described in the method.

Table 1. RC must be defined in the table legend. Subcellular must be changed to subcellular localization.

Put (aa) in front of Size of peptide and remove aa from the numbers under that column. Change signal peptide to position of signal peptide (aa) and remove aa from actual numbers. Elaborate how effector probability was calculated in the table legend.

I have modified it in the manuscript. But  the predicted value of the effector is difficult to describe in detail the intermediate process, because it is based on machine learning.  I have supplemented this content in my manuscript.

“EffectorP predicts fungal effectors based on machine learning.  Features that discriminate fungal effectors from secreted noneffectors are predominantly sequence length, molecular weight and protein net charge, as well as cysteine, serine and tryptophan content.”

Line 253. Italicize F. lateritium.

I have modified it in the manuscript.

Fug S2. It is not clear what INF1 is. It is a necrosis inducing effector from a pathogen. Please elaborate both in materials and methods and in the figure legend.

I have supplemented this content.

Line 229- The number of endophytic fungi colonizing in , what is the number of colonization? May be amount?

This is a comparison of relative quantities, no definite quantities are counted.

Line 305-306- which presumably related to the need of endophytes to maintain their population in plants. Need rewording.

I have modified it in the manuscript.

Line 341- that knockout of FlSp1 could help the fungus, I suppose it was the mutant but the WT that inhibits ROS burst and thus keeping immunity at a level that is suppressive for the fungus but prohibiting the occurrence of cell death.

This error is in the discussion section, but I have completely rewritten the discussion section.

Line 263-  However, why endophytes do not cause obvious disease symptoms. End this with a question mark.

This error is in the discussion section, but I have completely rewritten the discussion section.

Line 369-370-pathogenic fungi in hosts may be the different plant immune defense responses induced by effectors. It is violating what authors reported that strength of ROS is associated with the function of candidate effector. The qRTPCR results also suggested that plant defense genes known for pathogenic defense are involved. I encourage authors focus on interpreting the results obtained from study rather than citing a new hypothesis not backed up by the results.

This error is in the discussion section, but I have completely rewritten the discussion section.

Line 398-404- Same issue, a new hypothesis is opened with no link to the results of this study.

This section is in the discussion section, but I have completely rewritten the discussion section

Line 406-408- Need re-wording. It is not clear if ROS is contributing or inhibiting the disease.

This error is in the discussion section, but I have completely rewritten the discussion section.

Discussion general comments:

I suggest authors focuses further off on the results of this study. The discussion must be more specific and point-to-point connection of results to the research hypotheses and previous research. For example, author never explained why FLSp1 knock out improved resistance to R. solanacearum. They also did not explain immune pathways tested indirectly using qRTPCR of hallmark genes. The author suggested that the pathway is unknown while they could have used the results to make some inference on the pathways. There is no conclusion and proposition on the future studies.

Thank you very much for giving me such valuable comments, I have thought carefully about these issues and it has been very enlightening for me. I have therefore rewritten my discussion section in the manuscript. In the revised manuscript, we have discussed the function of FlSp1 based on experimental results. FlSp1 may contribute to the symbiotic relationship between fungi and plants to balance the immune system of plants.  And we preliminarily reveal the signal pathways that may be affected by FlSp1.

Reviewer 3 Report

Dear Authors,

Your paper presents a novel and interesting finding that FlSp1 is an effector that modulates plant immunity and oxidative stress in both positive and negative ways, depending on the context. The paper provides new insights into how endophytes balance their relationship with their host plants by modulating their immune responses and oxidative stress levels. The paper shows that FlSp1 is a novel effector that can trigger plant immunity by increasing reactive oxygen species (ROS) production, but also suppress it by reducing ROS accumulation and enhancing bacterial wilt resistance. The paper claims that its results provide a preliminary theoretical basis for F. lateritium-plant interactions and the mechanism by which F. lateritium helps plants to resist disease, but it does not explain how this is achieved or what are the implications for practical applications.The paper uses a combination of molecular, biochemical and physiological methods to characterize FlSp1 and its effects on the host plant and other pathogens. The methods are well-described and appropriate for the research question, and provides sufficient experimental evidence and data analysis to support its claims and hypotheses.

The paper could improve its clarity and readability by avoiding long and complex sentences, using more transitions and connectors, and avoiding grammatical errors. The paper could improve its clarity and readability by using more specific and consistent terminology, such as “F. lateritium colonization” instead of “fungal colonization”, and “plant immune system” instead of “plant immune defense”.

The paper could also improve its significance and impact by discussing how FlSp1 compares to other known fungal effectors, how it affects other endophytic or pathogenic interactions, and what are the implications for plant health and disease management.

The paper could also improve its significance and impact by citing more relevant literature to support its arguments and to compare and contrast its findings with previous studies on similar topics.

Overall, the manuscript is in pretty good shape, and reads fairly well. Hence, I recommend publishing of the manuscript with very minor adjustments.

Author Response

Dear Reviewers 3,

Thank you very much for taking the time to review the manuscript and for your very inspiring comments on its merits. All the opinions you give are valuable and very helpful for revising and improving our paper, as well as the important guiding significance to our researches. We have studied comments carefully and have made correction which we hope meet with approval. We start by retyping your comment in black font to facilitate discussion, and then present our response to the comment in red font.

Yours sincerely, Jianming Huang

Comment 1: The paper could improve its clarity and readability by avoiding long and complex sentences, using more transitions and connectors, and avoiding grammatical errors. The paper could improve its clarity and readability by using more specific and consistent terminology, such as “F. lateritium colonization” instead of “fungal colonization”, and “plant immune system” instead of “plant immune defense”..

Response 1: Thank you for your detailed review. We carefully proofread the manuscript and corrected grammatical and spelling errors in it. In addition, we have corrected those non-standard words in the manuscript.

Comment 2: The paper could also improve its significance and impact by discussing how FlSp1 compares to other known fungal effectors, how it affects other endophytic or pathogenic interactions, and what are the implications for plant health and disease management.

Response 2: Thank you very much for giving me such valuable comments, I have thought carefully about these issues and it has been very enlightening for me. I have therefore rewritten my discussion section in the manuscript. In the revised manuscript, we discussed the possible impact of FlSp1 on the endophytic relationship between F. lateritium and host plants, as well as its contribution to the balance of the plant immune system.

Comment 3: The paper could also improve its significance and impact by citing more relevant literature to support its arguments and to compare and contrast its findings with previous studies on similar topics.

Response 3: In the revised manuscript, we discussed the difference between the pathogen effector as an elicitor and FlSp1, which limits the excessive colonization of F. lateritium, and strives to avoid touching sensitive "nerves" of the plant immune system.

 Thank you again for your valuable comments!

Round 2

Reviewer 2 Report

Thanks for the revision. I suggested in my first review that authors use English proofing services. It looks like that that comment is ignored. According to the authors response, only my soft edits have been corrected. I encourage the authors to take the time and use help to improve the manuscript English.

Author Response

Dear Reviewer 2:

Thank you for your review of the paper and the valuable comments you have given me. I have carefully checked the paper and sent it to the "MDPI English Editing Service" for comprehensive English grammar modification. We have made correction carefully which we hope meet with approval.

Thank you again.

Yours sincerely, Jianming Huang
